# Re-testing as a method of implementing external quality assessment program for COVID-19 real time PCR testing in Uganda

Erick Jacob Okek[1,2,3]*, Fredrick Joshua Masembe[2], Jocelyn Kiconco[1], John Kayiwa[1], Esther Amwine[1,3], Daniel Obote[2], Stephen Alele[2], Charles Nahabwe[5], Jackson Were[4], Bernard Bagaya[2], Stephen Balinandi[1,3], Julius Lutwama[1,2,3], Pontiano Kaleebu[1,2,3]

1 Department of Arbovirology, Uganda Virus Research Institute, Entebbe, Uganda, 2 College of Health Sciences, Makerere University, Kampala, Uganda, 3 Viral Hemorrhagic Fevers Laboratory, Uganda Virus Research Institute, Entebbe, Uganda, 4 Department of Diagnostics, Mulago National Referral Hospital, Kampala, Uganda, 5 Department of Quality Assurance, Allied Health Professional Council, Kampala, Uganda

* okekerick@yahoo.com

**Data Availability Statement:** Data will be availed once the manuscript is accepted for publication.

**Funding:** Government of the republic of Uganda sends funds on a quarterly basis to support

## Abstract

### Background

Significant milestones have been made in the development of COVID19 diagnostics Technologies. Government of the republic of Uganda and the line Ministry of Health mandated Uganda Virus Research Institute to ensure quality of COVID19 diagnostics. Re-testing was one of the methods initiated by the UVRI to implement External Quality assessment of COVID19 molecular diagnostics.

### Method

participating laboratories were required by UVRI to submit their already tested and archived nasopharyngeal samples and corresponding meta data. These were then re-tested at UVRI using the WHO Berlin protocol, the UVRI results were compared to those of the primary testing laboratories in order to ascertain performance agreement for the qualitative & quantitative results obtained. Ms Excel window 12 and GraphPad prism ver 15 was used in the analysis. Bar graphs, pie charts and line graphs were used to compare performance agreement between the reference Laboratory and primary testing Laboratories.

### Results

**Eleven (11)** Ministry of Health/Uganda Virus Research Institute COVID19 accredited laboratories participated in the re-testing of quality control samples. 5/11 (45%) of the primary testing laboratories had 100% performance agreement with that of the National Reference Laboratory for the final test result. Even where there was concordance in the final test outcome (negative or positive) between UVRI and primary testing laboratories, there were still differences in CT values. The differences in the Cycle Threshold (CT) values were insignificant except for Tenna & Pharma Laboratory and the UVRI(p = 0.0296). The difference in the CT values were not skewed to either the National reference Laboratory(UVRI) or the

program activities. All authors of this manuscript are government employees paid wages at a periodic interval. For this project and genomic sequencing, CDC provided reagents in kind. There was no direct funding for this work and its publication.

**Competing interests:** No authors have any competing interest

primary testing laboratory but varied from one laboratory to another. In the remaining 6/11 (55%) laboratories where there were discrepancies in the aggregate test results, only samples initially tested and reported as positive by the primary laboratories were tested and found to be false positives by the UVRI COVID19 National Reference Laboratory.

## Conclusion

False positives were detected from public, private not for profit and private testing laboratories in almost equal proportion. There is need for standardization of molecular testing platforms in Uganda. There is also urgent need to improve on the Laboratory quality management systems of the molecular testing laboratories in order to minimize such discrepancies.

## Introduction

Laboratory External Quality Assurance program (EQA) are intended to ensure quality, timely and accurate results are released from the testing laboratories with ultimate goal of improving quality of patient care [1]. COVID19 diagnostics technology has evolved relatively fast since disease outbreak in 2019 [2]. This technological advancement is also associated with a number of errors that requires stringent regulation and monitoring of the accuracy, reliability, and precision of new testing technologies [3]. One such approach is to institute a reliable, efficient, and consistent External Quality Assessment program, with proper and timely root cause analysis conducted, corrective actions taken for non-conforming laboratories [4].

In Uganda, the antigen based Rapid Diagnostics test kits (AgRDTs) and Polymerase Chain Reaction (mainly Real time PCR) testing platforms have been validated and approved for use [5]. Currently, there are 67 Ministry of Health approved laboratories across Uganda conducting COVID19 PCR testing. The Uganda Virus Research Institute COVID19 National Reference Laboratory and the National Quality assurance committee has approved and periodically conducts three types of EQA: Proficiency Testing (PT) panel, re-testing/re-checking of quality control samples and on-site supervision. In the PT panel program, UVRI scientists calculate, prepare, concentrate, pool, aliquot, packages and distribute the panels to MoH approved laboratories. Laboratories conduct test, relay results to UVRI where sorting, analysis, report writing, dissemination of findings to relevant stake holders is done.

In the re-testing/re-checking, technical experts are sent to laboratories to randomly identify positive and negative samples from the archives and biobanks of primary testing laboratories, package and return with them. At the UVRI, re-testing is done, test outcomes are compared with those of the primary laboratories; reports are written and shared with relevant stakeholders. In both EQA methods, root cause analysis for under-performing laboratories is done & corrective actions taken. On-site supervision which is the 3rd EQA method is usually implemented alongside proficiency testing panel and re-testing.

When a COVID19 PCR result became a travel requirement in Uganda, the demand for the test overwhelmingly increased to a point where public facilities could not manage the workload. Subsequently, the Government apportioned part of this task to the private sector but maintained the oversight roles. Majority of these Laboratories were set up, assessed, and approved for operation in a rush due to national and international demands; moreover, molecular diagnostics is still relatively a new testing method in Uganda and many African countries.

To ensure test results reliability and continued improvement of Laboratory Quality management System, there was an urgent need to strengthen External quality assurance programs. This study evaluated the effectiveness of re-testing as a method of External quality assessment for COVID19 testing.

## Materials and methods

### Selection of the assessment and activation team

A team of experts were selected from the UVRI arbovirology Laboratory and dispatched to the various Laboratories across the country to retrieve samples for re-testing as part of quality control process. The National COVID19 Quality Assurance committee selected, vetted, and appointed a COVID19 National Laboratory Assessment and activation team that is composed of experts in molecular biology, policy & guidelines development, Laboratory Quality Management Systems (LQMS), and representative from the Allied Health Professional Council (diagnostics regulatory arm of the Ministry of Health). These experts were required to conduct quarterly visits (an interval of three months) of the approved testing sites. Amongst key activities are mentorship, supervision, vertical audit, and regulatory inspection.

### Participating health facilities and laboratories

A formal communication from the Ugandan Ministry of Health and Uganda Virus Research Institute was sent to all the 67 Ministry of Health assessed and approved COVID19 molecular testing facilities in Uganda prior to the visit by experts. Thirty-three (33) of the Laboratories had not been archiving (storing) samples and were excluded from participation. Twenty-one (21) of the remaining Thirty-four (34) did not have sufficient accompanying meta data (clients' demographics and clinical information) and were also excluded. Part of the aliquots from the positive samples retrieved were also used for genomic sequencing. Clients'/patients' details are crucial to making statistical sense and interpretation of the test results. Of the remaining thirteen (13) sites, two submitted insufficient sample volumes and were rejected according to the UVRI Arbovirology rejection criteria [6]. A total of Eleven(11) Laboratories qualified for inclusion in the program and these were: Mulago National Referral Hospital (Public), Examina Medical Laboratory(private), MAIA medicals(private), Test & Fly Laboratory (private), Kabale Regional Referral Hospital (public), Same day Laboratory (private), Bwindi Community Hospital(private not for profit),Medsafe Hospital (private), Gulu University multifunctional Laboratory (Teaching & research), Safari Laboratory (Private) and Tenna& Pharma (private).

### Sample and metadata retrieval from the primary testing laboratory

These experts were given cool boxes stocked with ice packs, thermometers for temperature monitoring, absorbent materials, and cotton wool. Public and University Laboratories such as Gulu University, Kabale Regional Referral Hospital and Mulago National Referral Hospital have freezers provided by Government and were able to store their samples at -20˚C. The rest of the private facilities had small freezers but still managed to archive samples at -20˚C. However, we noted that due to limited storage space and high-test positivity rates at the time, some samples could have been stored under inappropriate temperatures and other undesirable storage conditions, though we do not have proof of this. On reaching the facility, UVRI scientists interacted with the laboratory technicians (quality officers) who provided a list of all positive and negative nasopharyngeal samples in their biobanks and archives for the past three (3) months. About 1ml of the liquid aliquots were pipetted into cryovial tubes from each participating Laboratory and these were the quality control samples, part of the aliquot was separately

kept for genomic sequencing. Labels on the samples were cross checked for match with the duplicate sample identification in either the electronic database or in the records books. For laboratories with not more than ten(10)PCR positive COVID19 samples, all of them were picked for re-testing. For those with more than 10 positive samples, a probability sampling was done. In this, the total number of samples were counted and divided by a number that gave a convenient interval of selection. The same was done for negative samples. For positives, a total of 10 samples were selected (though there were instances where more were selected), while for negatives, a total of 20 samples were selected. Only samples stored for not more than three(3) months were considered for re-testing.

## Packaging, documentation & transportation

Selected samples were triple packaged according to a procedure described by *Karthik K et al.*; *2020* [7]. But briefly, selected nasopharyngeal samples were put in the primary nasopharyngeal container, then inside a zip-lock bag (secondary container) and finally in a cool box (tertiary container) with contents as previously discussed. A desiccant was put inside the zip-lock bag to absorb any moisture. Cotton wool was put between the ice packs and zip lock bags inside the cool box in order to avoid effects of moistures on the samples. For the meta data, the following information were captured; sample ID, testing facility, final test result, sample collection date, sample type, Cycle Threshold value (CT value) at different gene targets. Clinical details such as presenting signs and symptoms, disease severity among others were documented in the Laboratory Investigation form. Packaged samples with accompanying metadata were properly put inside the vehicles and transported to Uganda Virus Research Institute. Temporary storage at appropriate temperature were done prior to testing.

## Re-testing of nasopharyngeal samples at the UVRI

**TaqMan real time PCR was use.** *Principle of detection and amplification.* TaqMan real time PCR contains a set of forward and reverse primers along with probes that can bind the DNA/RNA between the binding sites for the primers [8]. The probe contains a fluorescence reporter molecule on its 5'end and a non-fluorescent quencher molecule on its 3' end. When the probe is intact, fluorescence is not present. During the amplification, the polymerase cleaves the fluorescence molecule from the probe producing fluorescence. The increase in fluorescence occurs only when the target sequence is complimentary to the probe and is amplified during the PCR reaction. Any non-specific amplification will not be detected due to the requirements for cleavage.

*Equipment preparation.* All work surfaces, pipettes, centrifuges, and other equipment were cleaned and decontaminated prior to use. Decontamination agents used included 5% bleach, 70% ethanol, and DNAzap™ and RNase AWA™ to minimize the risk of nucleic acid contamination.

*Nucleic acid extraction.* QIAamp Virus RNA Mini Kit was utilized in the RNA extraction: A method by *Liu Y; 2020* [9] was used to extract the RNA; but briefly, we aliquoted 140μL of nasopharyngeal specimen referred from primary testing sites for quality control purposes and eluted with 60 μL of buffer AVE. SARS-CoV-2 Negative Control in this kit was also extracted with the same protocol as for specimens. The Internal Control in the kit was added into the extraction mixture with 1μl/test to monitor the whole process. Manufacturer's recommended procedures (except as noted in recommendations above) were followed for sample extraction.

*Assay setup.* A method by *Shen M et al. 2020* [10] was used to set up the **reaction master mix, but briefly**; negative control and positive Control were included in each run. In the reagent setup room clean hood, Super Mix and RT-PCR Enzyme were mixed on ice or cold

block, this was meant to keep cold temperatures during preparations and use. The Super mix was thawed prior to use, subsequently, super mix and RT-PCR was mixed with enzyme mix by inversion 5 times or until the technician feels mixing was adequate. In the next step, centrifuge super mix and RT-PCR Enzyme were allowed to mix for 5 seconds, contents were collected at the bottom of the tube, and then the tube containing the mixture was placed in a cold rack. The number of reactions (N) to be set up per assay was then determined. In order to cater for possible pipetting error, excess reaction mix for the Negative Control, Positive Control, were made. After addition of the reagents, reaction mixtures were well agitated using vortex mixer. The mixture was centrifuged for 5 seconds and contents collected at the bottom of the tube, and then the tubes were placed in a cold rack. Reaction plates were set up in a 96-well cooler rack. 20 μL of master mix was after dispensed into each PCR tube. The entire reaction plate was covered, and the reaction plate was moved to the specimen nucleic acid handling area.

*Template addition*. Nucleic acid sample including positive and extracted negative control tubes were gently vortexed for approximately 5 seconds. Centrifugation was done for 5 seconds in order to collect contents at the bottom of the tube, and then the mixture in the tube was placed in a cold rack. After centrifugation, nucleic acid samples including positive and negative control were placed in tubes in the cold rack. Carefully, 5.0 μL of sample including positive and negative control were pipetted in each well. Other sample wells were covered during addition. Tips were changed after each addition. The column to which the sample has been added was securely capped to prevent cross contamination and to ensure sample tracking. Gloves were changed often and when necessary to avoid contamination.

## Creation and running of the PCR on the Applied Biosystems 7500 real time PCR instrument

Applied biosystems 7500 real time PCR instrument was launched, a new window created, and new experiment was chosen. Experimental properties were selected, after which targets and samples were selected. Whilst UVRI has many testing platforms, we opted to use Applied Biosystems in order to ensure uniformity and consistency in result analysis and interpretation.

## Testing platforms for COVID19 detection used by different laboratories across Uganda

A total of eight different molecular detection platforms were in use for COVID19 detection by participating in country laboratories. Two are closed systems (GeneXpert and U-STAR) while the others are open (Table 1). A Close system requires utilization of reagents or cartridges recommended by the manufacturer, while an open PCR gives options for utilization of any other reagents that can be compatible with the equipment. Being a national reference laboratory, UVRI has four testing platforms (Applied Biosystems, Quant Studio and Biorad) (Table 1). All these testing platforms were validated by the UVRI and Ministry of Health using a clearly defined Standard operating procedures. During re-testing, UVRI used only the Applied Biosystems platform in Berlin protocol, this was meant to ensure uniformity and reduce margin of error that could arise from inter platform differences.

## Genes and proteins on SARS CoV2 detected and reported by PCR testing platforms across different laboratories in Uganda

The majority of PCR testing platforms used by these laboratories detect genes while a few amplify viral proteins using different techniques but similar principles. The genes commonly detected are ORF1, E-gene, and N-gene. Bioer Fluorescence detection system used by MAIA

**Table 1. Real time PCR testing platforms being used for SARS CoV2 detection by various Ugandan laboratories.**

| Facility name | RT-PCR testing platform | Comment |
|---|---|---|
| Mulago NRH | 16 module GeneXpert | Closed system |
| Kabale RRH | 16 module GeneXpert | Closed System |
| Gulu University multifunctional Lab | Bioer Lineage System | Open System |
| Bwindi Community Hospital | Magnetic Induction Cycler | Open system |
| MAIA group of Laboratories | Bioer Fluorescent Detection System | Open system |
| Test & Fly Laboratory | Rotorgene | Open system |
| EXAMINA diagnostics Laboratory | U-STAR Technologies | Closed system |
| Tenna & Pharma Laboratory | SLAN 96 P Real time PCR | Open System |
| Same Day Laboratory | Biorad CFX 96 | Open System |
| Medsafe Hospital LTD | Quant Studio 5 | Open System |
| Safari Laboratory | Biorad CFX 96 | Open System |
| Uganda Virus Research Institute | Applied Biosystems, Quant studio 7, Biorad CFX and GeneXpert | A mixture of close and open systems |

Laboratories detected ORF1ab genes through the FAM channel while the same platform amplifies the N gene target sequence through the ROX channel (Fig 1). FAM is an important synthetic equivalent of a fluorescein dye used in oligonucleotide synthesis and molecular Biology. Bioer platform also detects the internal control (IC) in the VIC channel. VIC is a green color proprietary dye used to fluorescently label oligonucleotide at the 5'-end. The rest of the platforms either directly detected the genes or used other techniques beyond the scope of this study. N-gene was the most predominantly used for COVID19 detection by PCR testing platforms in Ugandan laboratories (32%), followed by ORF1 (32%), E-gene (18%). Only 14% of the participating laboratories ran and reported Internal Control before analyzing actual

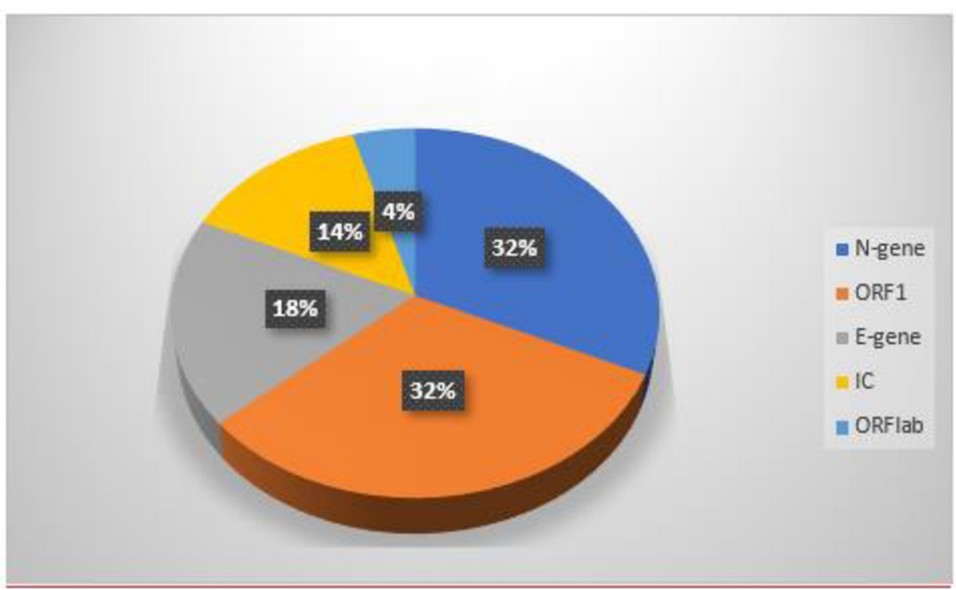

**Fig 1. Targets and genes reported by platforms of various testing laboratories in Uganda. Key:** ORF1 = open reading frame one. E-gene = Envelope gene. N-gene = Nucleocapsid gene. IC = Internal Control. ORF1ab = Open Reading Frame 1 ab.

samples (Fig 1). It is a requirement by the laboratory quality management system that Internal Controls should be run and documented to have passed before analysis of actual patients' or clients' samples. Utilization of ORF1ab gene for COVID19 detection was at 4%. All testing platforms detected and amplified at least two gene targets before confirming a positive test. In most platforms, a positive test was confirmed upon detection and amplification of N-gene and ORF1, while some platforms detected and amplified E-gene and ORF-1.

## Statistical methods

Cycle Threshold values of the quality control samples re-tested by the UVRI were exported into the CSV file facility by facility. The same identification number assigned to the sample by the primary testing laboratory was also assigned by UVRI during re-testing. At the UVRI, CT values were rounded off to two decimal places. Here a test is reported as Negative if CT value at two genes or targets exceeds 38.5.

To determine level of performance agreement, results from the primary laboratories were aligned with that of UVRI, sample by sample; matching was done by sample identification number. A result was reported as discordant when test result from the reference Laboratory does not agree with that of the primary testing laboratory and was reported as concordant when results from UVRI agrees with that of the primary testing Laboratory.

To perform a deeper analysis, the results were trimmed and exported to GraphPad prism (version 8). Line graphs of CT values of N-gene from UVRI were compared with that of the primary testing laboratory, sample by sample. Analysis of Variance and its P values in Graph Pad prism (version 8) were used to derive if there was any significant difference. Pie charts were drawn in Excel to determine utilization of different genes in RNA amplification and detection. To graphically display performance agreement, bar graphs were plotted in Excel spread sheet.

## Results

### Performance agreement between UVRI COVID19 national reference laboratory and the primary testing laboratories

A total of eleven(11) Ministry of Health approved COVID19 PCR testing Laboratories were considered for this study. These were the ones that submitted full metadata, sufficient sample volumes, had evidence of sample storage at the recommended temperature and presented well labelled quality control samples. Overall, 5/11 laboratories (45%) had a 100% performance agreement with UVRI, while the other 6/11 (55%) had varied number of discrepant results with the National Reference Laboratory (Fig 2). Out of the participating laboratories, two were public (Mulago National Referral Hospital and Kabale Regional Referral Hospital), seven were private for profit (Examina, MAIA, same day, Test & fly, Medsafe Hospital, Safari, Tenna & Pharma), one was private not for profit (Bwindi Community Hospital) and one was University Laboratory (Gulu University). Eight(8)of the Laboratories are located within the Kampala metropolitan area while two are located in Southwestern Uganda (Kabale Regional Referral Hospital and Bwindi Community Hospital) and one in Northern Uganda (Gulu University multifunctional Laboratory). Mulago National Referral Hospital and Gulu University had the highest number of discrepant results; three (3) samples from each of the Laboratories re-tested by the UVRI had negative results when they were initially tested positive. This was followed by Kabale Regional referral Hospital, Same day Laboratory, Safari Laboratory, and Bwindi Community Hospital with each having one discrepant result (false positive) upon re-testing by the National reference Laboratory (Fig 2). All samples initially tested as negative by the primary laboratories also tested negative by the UVRI and thus were excluded from both this plot and the overall analysis. Being

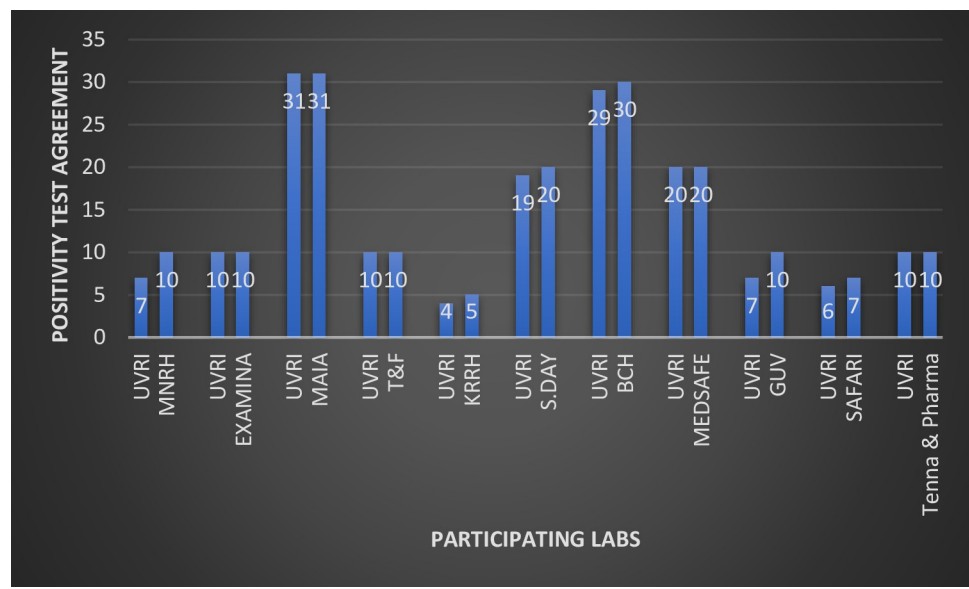

**Fig 2. Performance agreement between UVRI and initial testing laboratories. Key:** UVRI:Uganda Virus Research Institute. Examina: Examina Medical Laboratory. MAIA: MAIA medicals. T & F: Test and Fly Laboratory. KRRH: Kabale Regional Referral Hospital. S.Day: Same Day Laboratory. BCH: Bwindi Community Hospital. M.SAFE: Medsafe Hospital Limited. GUV: Gulu University multifunctional Laboratory.

a National Reference Laboratory, test outcome from the UVRI was treated as the correct and right result and conclusion of false result (positive or negative) was based on it.

## CT values of N genes of samples with discrepant results between UVRI and primary testing laboratories

For quality control samples with discrepant results between National Reference Laboratory and primary testing laboratories, we further triangulated the aggregate result by conducting in depth analysis of the CT values. For Mulago NRH, the CT values of the three discrepant results were still within the acceptable range for a true positive COVID19 PCR test by the UVRI. It is disturbing that samples with CT values of 31.5, 34.9 and 31.4 re-tested negative at the UVRI national reference laboratory whose cut off CT is 38.5 (Table 2). Similar findings were made for Gulu University multifunctional Laboratory where three samples with CT values of 18.79,18.84 and 28.27 that initially tested positive all turned negative upon retesting at the UVRI. For Kabale RRH (CT = 38.9), Bwindi Community Hospital(42.1), Same Day(40.1) and Safari Laboratory (35.5), CT values of N gene for samples with discordant results were out of range for acceptable values of a true positive COVID19 PCR test by the UVRI (Table 2).

## Fluctuation in CT values of N-gene between the primary testing laboratory and the UVRI COVID19 National Reference Laboratory

Cycle Threshold Values (CT) of N-genes of positive samples initially tested by the primary Laboratory were compared to that of the National Reference Laboratory at the Uganda Virus Research Institute. These were samples that tested positive by both the primary laboratory and the National Reference Laboratory; however, they had differences in the CT values. Among all samples tested, there was none with the exact CT value for the N gene at both the external site

**Table 2. Comparison of CT values of samples with discrepant results between the UVRI COVID19 National Reference Laboratory and the primary testing laboratories.**

| MNRH | UVRI | KRRH | UVRI | BCH | UVRI | S.day | UVRI | GUV | UVRI | Safari | UVRI |
|------|------|------|------|-----|------|-------|------|-----|------|--------|------|
| 31.15 | >38.5 | 38.9 | >38.5 | 42.1 | >38.5 | 40.1 | >38.5 | 18.79 | >38.5 | 35.5 | >38.5 |
| 34.9 | >38.5 | | | | | | | 18.84 | >38.5 | | |
| 31.4 | >38.5 | | | | | | | 28.27 | >38.5 | | |

Key

Column 1 & 2: CT values of the same samples tested by Mulago National Referral Hospital and Uganda Virus Research Institute but with discrepant results.

Column 3&4: CT values of the same sample but with discrepant results between Kabale Regional Referral Hospital and Uganda Virus Research Institute

Column 5&6: CT values of the same sample but with discrepant results between Bwindi Community Hospital and Uganda Virus Research Institute

Column 7&8: CT values of the same sample but with discrepant results between Same Day Laboratory and Uganda Virus Research Institute

Column 9&10: CT values of the same samples but with discrepant results between Gulu University and Uganda Virus Research Institute

Column 11 & 12: CT values of the same sample but with discrepant results between Safari Laboratory and Uganda Virus Research Institute

and reference Laboratory much as the differences were not statistically significant across board. The CT values for N gene was slightly higher (p = 0.2395) for all the samples re-tested at the UVRI compared to the initial testing Laboratory (Test & Fly Laboratory) (Fig 3A). For samples that tested positive by both Bwindi Community Hospital and Uganda Virus Research Institute, the differences in the CT value of N gene were tangling in between low and high for the two laboratories, much as Bwindi Community Hospital had higher CT values for the last two samples (p = .999) (Fig 3B). Quite a difference in the CT values of positive samples re-tested was observed between MAIA laboratory and UVRI National Reference Laboratory with MAIA reporting higher CT values across almost all samples (P = 0.0849) (Fig 3C). Much as it is not statistically significant (p = 0.2698), the CT values of N genes for results reported by UVRI were generally higher than that reported by Mulago National referral Hospital, the initial testing Laboratory (Fig 3D). For Gulu University and Safari Laboratories, the CT values for the N gene were tangling in between high and low (Fig 3E & 3F). The most interesting was that of UVRI and Tenna& pharma Laboratory. CT values for all the ten samples were significantly higher upon retesting by the UVRI (p = 0.0296) much as both laboratories agreed on the final test outcome (Fig 3).

## Discussion

This study intended to evaluate the effectiveness of re-testing as a method of implementation External Quality assessment of Ugandan molecular testing laboratories. It found some level of discrepancy of COVID19 PCR results between the external sites and the National Reference Laboratory at the UVRI. The discrepancies were spread across public and private facilities in near equal proportion. For example, Mulago National Referral Hospital which posted the highest number of discrepant results and Gulu University Multifunctional Laboratories are Public and University entities respectively. Kabale Regional Referral hospital is Government aided while Bwindi Community Hospital is a private not for profit supervised by the Uganda Protestant Medical Bureau (UPMB) while same Day Laboratory is a private facility. The discrepancies also cut across different platforms; example, Mulago NRH uses versant KPCR and gene Expert platforms, Kabale RRH uses GeneXpert platform, Bwindi Community Hospital uses GeneXpert and Magnetic induction Cycler, while same Day Laboratory uses U-star technologies and Biorad CFX.

Findings majorly point to variability in the inter-platform detection ranges. There could be some element of transmission and clerical errors that could have risen from heavy workload

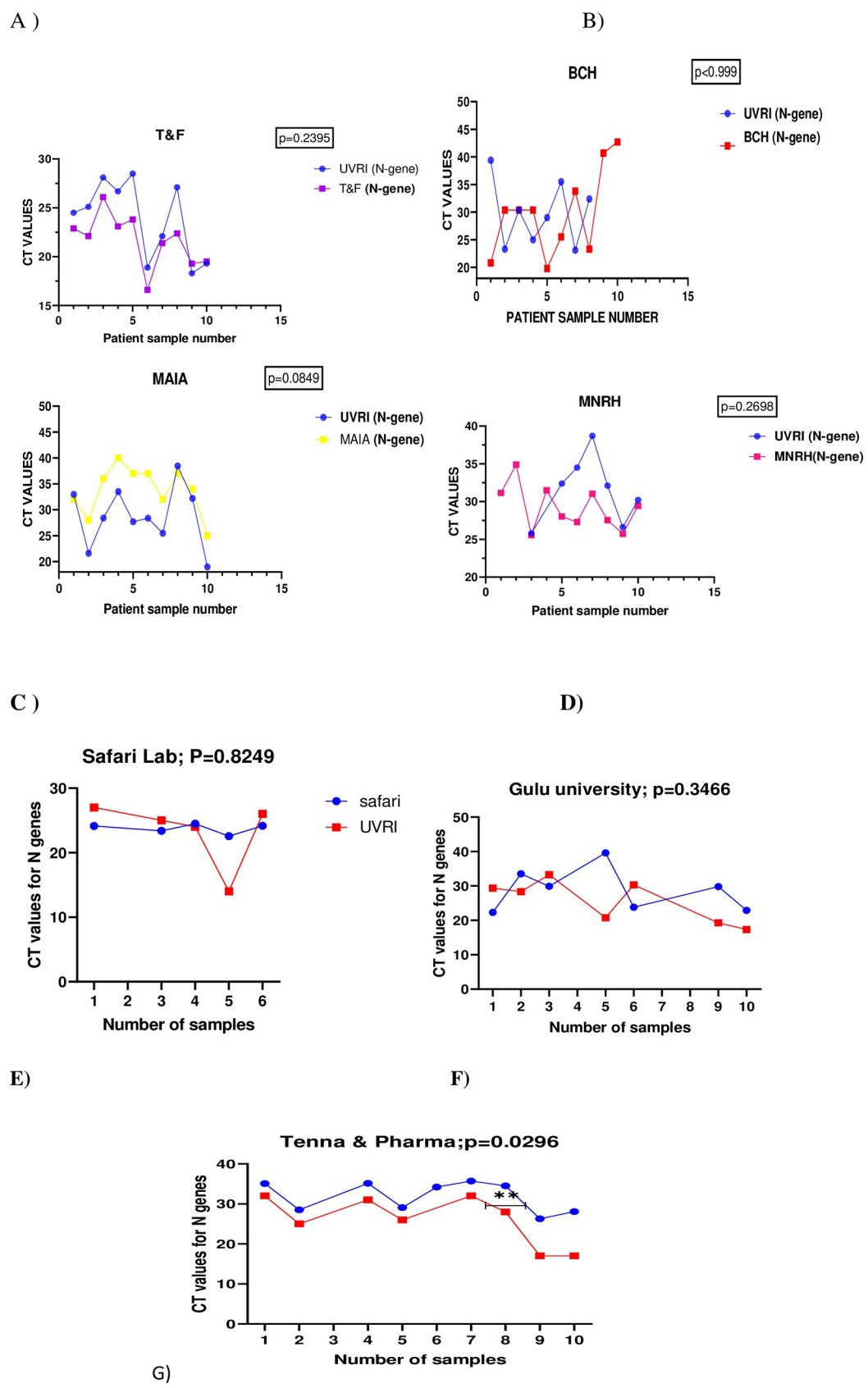

**Fig 3. Line graphs comparing CT values of N genes of quality control samples from different facilities and the UVRI COVID19 National Reference Laboratory.** 3A)CT value of Qc samples from Test & fly compared to UVRI, B)CT values of QC samples from Bwindi Community Hospital compared to UVRI, C) CT value of QC samples from MAIA compared to UVRI, D) CT values of QC samples from Mulago NRH compared to UVRI, E)CT values of QC samples from Safari Laboratory compared to that of UVRI, F)CT values of QC samples from Gulu University Laboratory compared to that of UVRI, G)CT values of QC samples from Tenna & Pharma Laboratory compared to that of UVRI.

and non-standardized reporting tools at that time. This study was conducted before the introduction of Laboratory Information Management systems and so the entire records and documentation processes were purely manual and error prone. According to the Ministry of Health Results Dispatch System (RDS), Mulago National Referral Hospital for example was testing over 500 samples per day at the peak of delta variant wave of COVID19. New staffs were just added, and yet molecular testing is highly sophisticated and requires advanced training, consistent practice, adequate and motivated work force. The majority of private facilities did not have sufficient capital to invest in molecular diagnostics given the unpredictability of COVID19 pandemics, especially from the Economic perspective. Unpublished report by the East Africa Community COVID19 Assessment common path committee found more than half of private COVID19 molecular diagnostics Laboratories in Uganda to be lacking in at least one of the 12 essential elements of a Laboratory Quality Management System. Because new Laboratories were being set up at a rapid speed, trained and competent personnel were being moved from one facility to another, leaving a huge gap of competent work force across most Laboratories. The interpretation of the CT values especially for open RT-PCR were very subjective. There was a high level of inter technician variability in the CT value interpretation from one Laboratory to another.

In order to understand the cause of the false positives, this study also compared actual CT values of N genes of quality control samples tested by the primary Laboratories and the UVRI COVID19 National Reference Laboratory and found positive samples from Mulago NRH and Gulu University which tested negative at the UVRI to have CT values within the acceptable range of a positive PCR result at the UVRI. These samples were concluded to have false positives according to guidance offered by the National Quality assurance committee for COVID19. This we believe could be errors arising from sample deterioration due to poor storage, inter-platform variability, packaging, and transportation. When poorly stored at wrong temperatures, RNA is very unstable and can deteriorate quickly. *J.Greenman et al.; 2015* reported that HIV RNA deteriorate quickly when dried blood spot is kept at room temperatures for long and so the viral copies reduced while CT values increases upon re-testing [11], however, we are not certain if this finding can be applicable to SARS CoV2 given the differences in properties and classification of the two viruses. A study done by *M Hardt et al. 2022* reported a significant reduction in detectable RNA in 75% of the swab solutions stored at 37˚C for 96 hours [12]. Most of these samples were stored for months at the primary testing laboratories before retrieval for quality control testing at the UVRI. Commercial reagent contamination and contamination in the laboratory workflow are among factors cited to cause false positives as in the case of Mulago NRH [13]. For the other three facilities with false positives, the CT values were out of range for a true positive test result according to the National reference Laboratory testing platform. This could be attributed to staff incompetency, clerical error, or transmission error. These factors have been reported to occur at the pre-analytical, analytical, and post-analytical phases of testing [14].

Even where there was perfect agreement between the initial testing laboratories and the National Reference Laboratory in the final test outcome, there were difference in the CT values though not statistically significant, except for Tenna & Pharma Laboratory. This could be due

to principles on which the different testing technologies were built on. All facilities whose results were compared to that of UVRI had different testing platforms with inter-technological differences. *Daniel Rhoads et al. 2021* reported that CT values can vary within and between methods [15]. The college of American pathologists surveyed over 700 laboratories using proficiency testing materials produced from the same batch and found CT values by different instruments to vary by as much as 14 cycles. Within a single gene target for a single method, up to 12 cycles were reported to have been seen across all laboratories.

## Conclusion

Discrepant COVID19 PCR results (false positives and false negatives) were caused by a wide range of factors; amongst which are clerical errors, inadequate storage facility, inter-differences across testing technologies, gaps in the chain of custody, huge workload, clerical, and transmission errors. Inter-laboratory comparison of results(re-testing) is an effective method of implementing External quality assurance program of molecular testing. Laboratories should invest more in developing quality management systems and enroll for accreditation. There should be continued investment in COVID19 and other molecular testing External Quality Assurance programs. Authorities in Uganda, and other Countries especially the line Ministry of Health can benchmark on these findings to expand the external quality assurance to other disease programs. Finally, it is not possible to reproduce the exact CT values for the same sample(s) run across different testing platforms. It is appropriate to give a range of acceptable values for a positive COVID19 PCR test.

## Limitation of the study

Whilst these samples were being stored at the right temperatures during retrievals, we cannot guarantee the same was being consistently done for the three months while the samples were at these external sites. Majority of these laboratories especially private ones do not have power back up and yet electricity black out is a common occurrence in Uganda. Fluctuation in temperatures can lead to RNA degradation and protein denaturation; inconsistency power supply can cause temperature fluctuations. This was a program related work, and we were unable to control for those confounding.

## Supporting information

**S1 Data.**
(XLSX)

## Acknowledgments

Let me extend my gratitude to the Ugandan Ministry of Health for putting up a spirited fight against different waves of COVID19 pandemic. Let me also applaud the Incident Management Team (IMT) and the Laboratory pillar of the Ministry of Health for the consorted efforts put against COVID19 and other emerging and re-emerging infections. Special gratitude goes to the Uganda Virus Research Institute (UVRI) and the National Quality assurance committee for the technical roles played. Much appreciation goes to Center for Diseases Control & prevention (CDC) under the candid leadership of Mr.Thomas Nsibambi for supplying reagents and other logistics that was used for running the QC samples. Finally, let me acknowledge COVID19 PCR testing laboratories for participating in the External Quality Assessment program.

## Author Contributions

**Conceptualization:** Erick Jacob Okek.

**Data curation:** Erick Jacob Okek, Daniel Obote, Stephen Alele, Charles Nahabwe, Jackson Were.

**Formal analysis:** Erick Jacob Okek, Fredrick Joshua Masembe, John Kayiwa.

**Funding acquisition:** Julius Lutwama, Pontiano Kaleebu.

**Investigation:** Erick Jacob Okek, Fredrick Joshua Masembe, Jocelyn Kiconco, John Kayiwa, Esther Amwine, Stephen Alele, Stephen Balinandi.

**Methodology:** Erick Jacob Okek, John Kayiwa, Esther Amwine, Daniel Obote, Stephen Alele.

**Project administration:** Jackson Were, Julius Lutwama.

**Resources:** Julius Lutwama, Pontiano Kaleebu.

**Supervision:** Bernard Bagaya, Julius Lutwama, Pontiano Kaleebu.

**Validation:** John Kayiwa, Charles Nahabwe, Stephen Balinandi.

**Visualization:** Jackson Were, Bernard Bagaya, Stephen Balinandi.

**Writing – original draft:** Erick Jacob Okek.

**Writing – review & editing:** Erick Jacob Okek, Bernard Bagaya, Stephen Balinandi, Julius Lutwama, Pontiano Kaleebu.

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
