## [Decision Letter · Decision Letter 0]

5 Jul 2023

PONE-D-23-16243RE-TESTING AS A METHOD OF IMPLEMENTING EXTERNAL QUALITY ASSESSMENT PROGRAMME FOR COVID-19 PCR TESTING IN UGANDAPLOS ONE

Dear Dr. Okek,

Thank you for submitting your manuscript to PLOS ONE. After careful consideration, we feel that it has merit but does not fully meet PLOS ONE’s publication criteria as it currently stands. Therefore, we invite you to submit a revised version of the manuscript that addresses the points raised during the review process.

All reviewers had concerns, some of which were major. Kindly review their comments and address as appropriate.

We look forward to receiving your revised manuscript.

Kind regards,

Chika Kingsley Onwuamah, Ph.D.

Academic Editor

PLOS ONE

 “This work did not have direct funding. It's part of routine program activities of the Uganda Virus Research Institute. The institute was designated as COVID19 National reference Laboratory by the Ugandan Ministry of Health. UVRI was also designated as regional referral laboratory for COVID19, influenza and viral hemorrhagic fevers by CDC and WHO. A lot of samples have been referred from South Sudan, Democratic Republic of Congo, Burundi amongst others. As part of the dissemination plan, staffs are encouraged to publish in peer reviewed journals.”

d) If you did not receive any funding for this study, please state: “The authors received no specific funding for this work.

4. Please amend the manuscript submission data (via Edit Submission) to include authors Jocelyn Kiconco, John Kayiwa, Esther Amwine, Daniel Obote, Stephen Alele, Charles Nahabwe, Jackson Were, Bagaya Bernard, Balinandi Stephen, Thomas Nsibambi, Julius Lutwama and Pontiano Kaleebu.

Reviewers' comments:

Reviewer's Responses to Questions

**Comments to the Author**

1. Is the manuscript technically sound, and do the data support the conclusions?

Reviewer #1: Yes

Reviewer #2: No

Reviewer #3: No

2. Has the statistical analysis been performed appropriately and rigorously? 

Reviewer #1: Yes

Reviewer #2: No

Reviewer #3: No

3. Have the authors made all data underlying the findings in their manuscript fully available?

Reviewer #1: No

Reviewer #2: No

Reviewer #3: Yes

4. Is the manuscript presented in an intelligible fashion and written in standard English?

Reviewer #1: No

Reviewer #2: Yes

Reviewer #3: No

5. Review Comments to the Author

Reviewer #1: REVIEW: RE-TESTING AS A METHOD OF IMPLEMENTING EXTERNAL QUALITY ASSESSMENT PROGRAMME FOR COVID-19 PCR TESTING IN UGANDA

1. What type of PCR? Is it conventional or real time PCR? This should be stated at required points in the paper.

2. There is need to check the use of the word “quadrille” and use as appropriate.

3. The statement starting from “In September …” needs to be referenced.

4. The paragraph starting from "A total of eight different genes, dyes …” is ambiguous and disjointed. It is important to state clearly the functons of the primers and the dyes. The dyes are not the targets.

5. Abbreviations are not expected at the beginning of a sentence as seen in the statement starting from “CT …”

6. In what form were the samples retrieved? As extracted DNA or in the crude form? What was the storage condition at the different facilities prior to retrieving the samples from them? How many days after initial testing were the samples re-tested?

7. Quotation referred to J. Greenman et. al. needs to be revised for clarity.

Reviewer #2: As a Quality Assurance measure during the rollout of SARS-CoV-2 diagnostic testing, Okek et al. performed confirmatory qRT-PCR testing after initial testing was done at multiple hospitals, private test centers, and university sites within Uganda. The confirmatory testing was performed at the Uganda Virus Research Institute and included re-testing of 10 positive samples and 20 negative samples from each external site. Among the 11 external sites that participated, 5 had fully concordant positive/negative results with UVRI. At 2 external sites, there were three false-positives each, and at 4 external sites, there were one false-positive each. There were no false-negatives. This study nicely illustrates the importance of centralized confirmatory testing as multiple assays were rolled out quickly during the SARS-CoV-2 pandemic. As suggested below, the manuscript would be strengthened by adding more details and reframing some interpretations.

- How were sites chosen to be offered confirmatory testing? How many sites were offered confirmatory testing and declined? Could there have been bias towards better-performing sites in those that agreed to undergo confirmatory testing?

- In the Discussion, the authors allude to RNA degradation as a potential reason for tests going from positive (at the initial external site) to negative (at UVRI). I agree this is a very likely explanation for the “false positives” observed, and the authors should include more details in the Methods and Results sections about the temperature and duration of storage for samples at each site. For the “false positive” samples, it would be important to evaluate whether their storage conditions were different from the others.

- Figure 1 – please label the y-axis. If the y-axis indicates the number of positive tests, then to me it looks like more positive tests were found at UVRI than the external sites, which would indicate false-negatives at the external sites, not false-positives.

- I don’t understand the relevance of the genes, dyes, and targets reported by different laboratories. What question(s) were the authors asking by evaluating this information? Instead, it would be more informative to provide a table listing the assays, kits, and machines used by each external site.

- As the authors note in the Discussion, it’s very difficult to compare Ct values between assays. This point should be acknowledged sooner, and while OK to present data about the Ct values (Table 2 and Figure 3), I think the interpretation needs to be much softer. For example, it is not accurate to expect that sample with a Ct of 31.5 tested at an external site should have had the same (or even a similar) Ct when tested at UVRI. Similarly, it may not be accurate to state that “For the other three facilities with false positives, the CT values were out of the acceptable range for a positive test result according to the National reference Laboratory,” since the cutoff needs to be determined for the specific assay and machine being used. I also don’t think it’s appropriate or necessary to compare Ct values by statistical testing, and the authors do not state what test was used.

- In Table 2, what do the different rows indicate? It would be more informative to list each sample on a separate row and label the rows.

- In Figure 3, why are only seven sites shown?

- I disagree with the authors that they found a high level of discrepancy. For one thing, it was very good that no false-negatives were found. The few false-positives that were found could be explained by storage, RNA degradation, and different assays used. The differences in Ct observed between sites/assays is expected. Although the exact Cts are not expected to be the same, you would generally expect the difference between any two assays to be consistent between samples, and looking at Figure 2 this seems to be the case.

- While human error could be a component of discrepant results, some of the wording is overstated e.g. “With the stated inadequacies, discrepant results were inevitable.” Furthermore, it’s difficult to invoke incompetency as an explanation without testing operator competency. This section should be rewritten to be more constructive.

Reviewer #3: Methodology

• What is the frequency of the visit of experts to the labs? Is it one-off or periodic?

• How long and at what temperature are samples stored before they are selected for testing?

• Was the method used for testing by the labs retrieved also?

• What is the “right” temperature at which retrieved samples are stored at the reference lab?

• Packaging, Documentation & transportation. Selected samples were triple packaged according to (7). This is an incomplete statement

Results

• The methodology is described in the result section as it had to do with Fig 2 (VIC, FAM, ROX, and CV5). This should be moved to the methodology section.

• Differences in CT values are expected even when the same samples are run in duplicates in the same run. There is a need to confirm the assays' intra and inter-assay variabilities to interpret CT values within the context of the manufacturer’s performance characteristics. It is important to know if the same RNA extraction method was used between the reference and primary labs in order for an objective comparison.

Discussion

• There is a need for specification with this statement “Mulago NRH for example was testing over 500 samples”. Is it 500 per day/month/year?

• The methods used for COVID-19 testing at some labs were mentioned in the discussion but this was not listed as part of the metadata obtained from the facilities along with the samples.

• It is unclear how the authors describe QMS to be lacking in some labs when the study did not assess its implementation.

• A mention that the samples were stored for months at the primary testing labs before collection for retesting negates the essence of the quality assurance re-testing program. Since the aim of this program is not to assess the quality of storage but of results, the procedure used was not suitable for the objective.

Conclusion

• Most of the factors listed as affecting the discrepancy in results were not investigated in this study hence, cannot be categorically attributed.

General

• Confounders need to be addressed before making any general statements else they should be listed as limitations.

• Considering the requirement for confidentiality, the names of the labs should have not been mentioned.

• A review by an English editor will provide some more clarity.

6. PLOS authors have the option to publish the peer review history of their article (what does this mean?). If published, this will include your full peer review and any attached files.

Reviewer #1: **Yes: **Babatunde Akeem Saka

Reviewer #2: No

Reviewer #3: No

---

## [Author Response · Author response to Decision Letter 0]

16 Aug 2023

Response to questions, comments and guidance offered by the Scientific reviewers on Manuscript titled “Re-testing as a method of implementing External Quality Assessment Program for COVID-19 real time PCR testing in uganda.”

Reviewer one

Qn one: What type of PCR?, is it conventional or Real Time PCR? This should be stated at required points in the paper.

Response: All testing platforms were real time PCR. Tittle in the revised manuscript has real time PCR added and also in the main text.

Qn Two: There is need to check the use of the word “quadrille” and use as appropriate.

Response: Thank you for this observation; the word quadrille has been removed in the revised manuscript and replaced by “overwhelmingly increased”

Qn three: The statement starting with “In September…” need to be referenced.

Response: The entire paragraph has been removed and replaced by a more befitting statement in the revised manuscript.

Qn four: The paragraph starting with “ A total of eight different gene dyes” is ambiguous and disjointed. It is important to clearly state the functions of the primers and probes . The dyes are not probes.

Response: I do appreciate your keen observation and clarification. That entire section has been re-written, and I believe the ambiguity is no more. The revised section clearly states the roles of the function of different probes and primers.

Qn five: Abbreviations are not expected at the beginning of a sentence as seen in the statement starting from “CT…”

Response: Thank you so much for this guidance. The revised paragraph now starts with Thermocycler Value as opposed to CT in the original version.

Qn Six: In what forms were the samples retrieved? As extracted DNA or in crude form? , What was the storage conditions at the different facilities prior to retrieving the samples from them? How many days after initial testing were the samples re-tested?

Response: These were aliquots of stored Nasopharyngeal swabs in crude form stored at Negative 20oC across all facilities. Samples for the past three months were collected.

Qn seven: Quotation referred to J.Greenman et al need to be revised for clarity.

Response: The citation has been clarified in the revised manuscript

Reviewer 2

Qn one: How were the sites chosen to be offered confirmatory testing? How many sites were offered confirmatory testing and declined? Could there have been biased towards better performing sites in those that agreed to undergo confirmatory testing?

Response: All the 67 sites were eligible, but they were excluded because they didn’t meet the inclusion criteria. The majority of them were not archiving samples while some did not have sufficient metadata, and few submitted insufficient sample volumes. Inclusion criteria has been well stated in the revised manuscript. 

Qn two: In the discussion, the authors allude to RNA degradation as a potential reason for test going from positive (at the initial external site) to negative (at UVRI). I agree this is a very likely explanation for the “false positive” observed, and the authors should include more details in the methods and result sections about the temperatures and duration of storage of samples at each site . For the “false positive” samples, it would be important to evaluate whether their storage conditions were different from others.

Response: An entire section has been created on storage conditions, temperature monitoring and duration has been included in the revised manuscript. It addresses all your queries. 

Qn three: Figure 1 – please label the y-axis. If the y-axis indicates the number of positive tests, then to me it looks like more positive tests were found at UVRI than the external sites, which would indicate false-negatives at the external sites, not false-positives.

Response: Y-axis has been in the revised manuscript. I think it still indicates a false positive by an external site because the positive issued to the client by an external site.

Qn four: I don’t understand the relevance of the genes, dyes, and targets reported by different laboratories. What question(s) were the authors asking by evaluating this information? Instead, it would be more informative to provide a table listing the assays, kits, and machines used by each external site.

Response: Thank you so much for this observation and guidance. A table listing the testing platform by each facility has been created in the revised manuscript. Information on genes, dyes and targets were put forward in order for the readers to appreciate the diversity in testing platforms in Uganda .

Qn five : As the authors note in the Discussion, it’s very difficult to compare Ct values between assays. This point should be acknowledged sooner, and while OK to present data about the Ct values (Table 2 and Figure 3), I think the interpretation needs to be much softer. For example, it is not accurate to expect that sample with a Ct of 31.5 tested at an external site should have had the same (or even a similar) Ct when tested at UVRI. Similarly, it may not be accurate to state that “For the other three facilities with false positives, the CT values were out of the acceptable range for a positive test result according to the National reference Laboratory,” since the cutoff needs to be determined for the specific assay and machine being used. I also don’t think it’s appropriate or necessary to compare Ct values by statistical testing, and the authors do not state what test was used.

Response: We take good note of your guidance and observation. As advised, the language has been softened in the revised manuscript. UVRI used the berlin protocol to test these samples; this protocol was WHO approved and so we treated it as gold standard. At the time, Uganda as a country wanted to set a cut off for acceptable positive test and this study was meant to be one of the key informers. With similar storage conditions as it was for the external sites, variation in temperature was some how controlled. We used ANOVA in graph pad prism to compare the CT values.

Qn six: In Table 2, what do the different rows indicate? It would be more informative to list each sample on a separate row and label the rows.

Response: We tried and found that listing each sample on a separate row and labelling will over crowd the work. We instead opted to put a key for each column just below the table.

Qn seven : In Figure 3, why are only seven sites shown?

Response: Drawing 11 graphs for 11 sites would be monotonous since findings were similar. We instead opted to consider regional balance, ownership (public vs private) in the plotting the graphs. The seven facilities were representative enough.

Qn Eight : I disagree with the authors that they found a high level of discrepancy. For one thing, it was very good that no false negatives were found. The few false positives that were found could be explained by storage, RNA degradation, and different assays used. The differences in Ct observed between sites/assays is expected. Although the exact CTs are not expected to be the same, you would generally expect the difference between any two assays to be consistent between samples and looking at Figure 2 this seems to be the case.

Response: Thank you for your in-depth analysis. In-ability to monitor storage conditions for a longer period has been stated as a major limitation to this study. However, we tried to control storage as a cofounding by excluding sites without freezers. In the discussion, we also categorically acknowledged RNA degradation as a possible factor and cited literatures that confirms it as well.

Qn Nine: While human error could be a component of discrepant results, some of the wording is overstated e.g. “With the stated inadequacies, discrepant results were inevitable.” Furthermore, it’s difficult to invoke incompetency as an explanation without testing operator competency. This section should be rewritten to be more constructive.

Response: Some of the overstated words have been changed in the revised manuscript. We also cited unpublished report from East Africa Community COVID19 common path assessment report that identified personal as a component deficient in most Ugandan molecular testing Laboratories. However, we do acknowledge that personal competency was not one of the outcomes of our study.

Reviewer 3

Qn one: What is the frequency of the visit of experts to the labs? Is it one-off or periodic?

Response: A section titled “Selection of assessment and activation team” has been included in the revised manuscript. It has all the details of the questions you have asked.

Qn two: How long and at what temperature are samples stored before they are selected for testing?

Response: A section on samples and meta data retrieval has been enriched to address most of the questions you have raised in the revised manuscript

Qn three: Was the method used for testing by the labs retrieved also?

Response: A table with testing platforms for different Laboratories has been created in the revised manuscript.

Qn Four: What is the “right” temperature at which retrieved samples are stored at the reference lab?

Response: UVRI temporarily stores samples at negative 80 degree Celsius for a short term (3 months) and negative 96oC for long storage.

Qn five: Packaging, Documentation & transportation. Selected samples were triple packaged according to (7). This is an incomplete statement.

Response: A full statement has been put in that section of the revised manuscript. It addressed the comment you have raised.

Qn Six: The methodology is described in the result section as it had to do with Fig 2 (VIC, FAM, ROX, and CV5). This should be moved to the methodology section.

Response: Thank you for this keen observation and guidance. The section has been moved from results section to method section in the revised manuscript.

Qn 7: Differences in CT values are expected even when the same samples are run in duplicates in the same run. There is a need to confirm the assays' intra and inter-assay variabilities to interpret CT values within the context of the manufacturer’s performance characteristics. It is important to know if the same RNA extraction method was used between the reference and primary labs in order for an objective comparison.

Response: Unfortunately, platforms and assays have different extraction methods. However, because UVRI uses berlin protocol (one of the earliest approved by WHO at the onset of COVID19), we expect all other testing platforms to agree with it much as inter assay differences is inevitable. 

Qn 8: There is a need for specification with this statement “Mulago NRH for example was testing over 500 samples”. Is it 500 per day/month/year?

Response: This statement has been clarified in the revised manuscript

Qn 9: The methods used for COVID-19 testing at some labs were mentioned in the discussion, but this was not listed as part of the metadata obtained from the facilities along with the samples.

Response: A separate table with list of testing platforms across various laboratories have been included in the revised manuscript. 

Qn 10: It is unclear how the authors describe QMS to be lacking in some labs when the study did not assess its implementation.

Response: In this study, we did not have sufficient proof to identify any inadequacy in the QMS. However, we based on the unpublished report by East Africa Community COVID19 common path assessment report that found some lacks to be deficient in the QMS. We have however revised the statement in the revised manuscript.

Qn 11: A mention that the samples were stored for months at the primary testing labs before collection for retesting negates the essence of the quality assurance re-testing program. Since the aim of this program is not to assess the quality of storage but of results, the procedure used was not suitable for the objective.

Response: In the revised manuscript, we stated that the samples were stored at -20oC at the external sites for at most three months, much as we did not have proof of consistency of the storage temperatures for all these months. 

Qn 12: Most of the factors listed as affecting the discrepancy in results were not investigated in this study, hence, cannot be categorically attributed.

Response: We actually did not attribute those factors in totality, but we only assumed or hypothesized. Subsequent studies will be designed to investigate those factors. 

Qn 13: Confounders need to be addressed before making any general statements else they should be listed as limitations.

Response: Thank you for the guidance; a section on limitation has been created. Some of the confounding such as storage has been listed amongst limitations.

Qn 14: Considering the requirement for confidentiality, the names of the labs should have not been mentioned.

Response: These Laboratories are public utility facilities. Besides, this work was more of program and regulatory as opposed to pure academic and research. Non the less, we have softened some of the tough languages used in the original manuscript which was some how implicative. 

Qn 15: A review by an English editor will provide some more clarity.

Response: This comment lack clarity; I believe the quality of English in the revised manuscript is much better now.

---

## [Decision Letter · Decision Letter 1]

14 Sep 2023

PONE-D-23-16243R1RE-TESTING AS A METHOD OF IMPLEMENTING EXTERNAL QUALITY ASSESSMENT PROGRAMME FOR COVID-19 REAL TIME PCR TESTING IN UGANDAPLOS ONE

Dear Dr. Okek,

Thank you for submitting your manuscript to PLOS ONE. After careful consideration, we feel that it has merit but does not fully meet PLOS ONE’s publication criteria as it currently stands. Therefore, we invite you to submit a revised version of the manuscript that addresses the points raised during the review process.

The reviewers acknowledge the manuscript has greatly improved. However, all the issues raised have not be addressed. Kindly address all issues raised by the three reviewers in both rounds of reviews.

We look forward to receiving your revised manuscript.

Kind regards,

Chika Kingsley Onwuamah, Ph.D.

Academic Editor

PLOS ONE

Additional Editor Comments:

Please, the issues raised by all three reviewers have not been fully addressed. Kindly address all satisfactorily to allow us proceed. See the reviewers comments included

Reviewers' comments:

Reviewer's Responses to Questions

**Comments to the Author**

1. If the authors have adequately addressed your comments raised in a previous round of review and you feel that this manuscript is now acceptable for publication, you may indicate that here to bypass the “Comments to the Author” section, enter your conflict of interest statement in the “Confidential to Editor” section, and submit your "Accept" recommendation.

Reviewer #1: (No Response)

Reviewer #2: (No Response)

2. Is the manuscript technically sound, and do the data support the conclusions?

Reviewer #1: Yes

Reviewer #2: No

3. Has the statistical analysis been performed appropriately and rigorously? 

Reviewer #1: Yes

Reviewer #2: No

4. Have the authors made all data underlying the findings in their manuscript fully available?

Reviewer #1: Yes

Reviewer #2: No

5. Is the manuscript presented in an intelligible fashion and written in standard English?

Reviewer #1: No

Reviewer #2: No

6. Review Comments to the Author

Reviewer #1: 1. Previous corrections such as the use of abbreviations at the beginning of a sentence is still in the manuscript.

2. The word quadrilled was still used wrongly in the manuscript.

3. CT value means cyclethreshold and should be referred to as such. Thermocycler value cannot be abbreviated as CT.

4. The use and interpretation of dyes and genes is still confusing in this manuscript. For clarity, dyes are the coloured probes used to detect the genes intended for identification (amplification). Dyes are not to be grouped as items to be identified as this paper implied especially summarised in the pie chart.

Reviewer #2: While the manuscript has improved, there are important comments from myself and other reviewers that have not been adequately addressed. I have some examples below, but I recommend that the authors carefully go through the prior detailed review to make sure they understand and address each comment. The manuscript contains valuable data, but there are still important inaccuracies in the analysis and interpretations. Just a couple of examples:

-A fundamental problem with the paper is the expectation that the specific Ct values produced by the reference lab should match the specific Ct values produced by each other lab. This is not true, because machines and assays have different ways of producing and measuring fluorescence, which are not expected to generate the same Ct value for the same sample. Here is an example of a study that demonstrates variable Ct values across assays and platforms, even though all are valid to use:

https://journals.asm.org/doi/10.1128/jcm.00821-20

Because of this well known inter-assay variability, specific comparisons are not appropriate, though the relative value of each sample can be compared e.g. using correlation tests.

-I still do not understand Figure 1. Using the external site MNRH as an example (first set of bars), it looks like 7 samples tested positive at MNRH and 10 samples tested positive at UVRI. This would mean that there were either 3 false-positives at UVRI or 3 false-negatives at MNRH. This is the opposite of what is described in the text.

-As reviewers 1 and 2 noted, the description of dyes and probes in the various real-time PCR assays is confusing and misleading. Dyes are not probes; each assay will use a dye, and some will also use a probe. By placing them in the same pie chart in Figure 2, the authors make it seem as though each assay uses either a dye or a probe.

-Again, these are just a few examples of where I do not think the authors have adequately addressed prior comments.

7. PLOS authors have the option to publish the peer review history of their article (what does this mean?). If published, this will include your full peer review and any attached files.

Reviewer #1: No

Reviewer #2: No

---

## [Author Response · Author response to Decision Letter 1]

29 Oct 2023

Response to Reviewers

Reviewer #1: 1. Previous corrections such as the use of abbreviations at the beginning of a sentence is still in the manuscript.

Response: Thank you for the guidance; abbreviations locally used such as Ministry of Health (MoH) and Uganda Virus Research Institute (UVRI) have been written in full in the abstract. However, there are universally accepted abbreviations such as COVID19, PCR that the target audience for this manuscript are conversant with. 

2. The word quadrilled was still used wrongly in the manuscript.

Response: Thank you for this observation. However, the word “quadrilled” does not exist in this revised manuscript to the best of my knowledge. Deliberate efforts have been put to avoid colloquial words in the revised manuscript. 

3. CT value means cyclethreshold and should be referred to as such. Thermocycler value cannot be abbreviated as CT.

Response: Thank you so much for this correction. What was previously written as Thermocycler value has been changed to cycle Threshold in the entire revised manuscript as guided

4. The use and interpretation of dyes and genes is still confusing in this manuscript. For clarity, dyes are the colored probes used to detect the genes intended for identification (amplification). Dyes are not to be grouped as items to be identified as this paper implied especially summarized in the pie chart.

We take note of your comment. The entire section of genes and dyes has been re-written in the revised manuscript. I believe the ambiguity has been clarified in the revised manuscript. 

Reviewer #2: While the manuscript has improved, there are important comments from myself and other reviewers that have not been adequately addressed. I have some examples below, but I recommend that the authors carefully go through the prior detailed review to make sure they understand and address each comment. The manuscript contains valuable data, but there are still important inaccuracies in the analysis and interpretations. Just a couple of examples:

-A fundamental problem with the paper is the expectation that the specific Ct values produced by the reference lab should match the specific Ct values produced by each other lab. This is not true, because machines and assays have different ways of producing and measuring fluorescence, which are not expected to generate the same Ct value for the same sample. Here is an example of a study that demonstrates variable Ct values across assays and platforms, even though all are valid to use:

https://journals.asm.org/doi/10.1128/jcm.00821-20

Because of this well-known inter-assay variability, specific comparisons are not appropriate, though the relative value of each sample can be compared e.g. using correlation tests.

Response: Thank you for this observation and comment. In the dissuasion of the revised manuscript, we acknowledge inter-assay differences as a possible cause of the discrepancies. Much as the exact values cannot be reproduced across different platform, there should be an acceptable range of values which are usually +/- 2SD. However, results from the UVRI are treated as the true results by the National Quality assurance committee after weighing in so many variables

-I still do not understand Figure 1. Using the external site MNRH as an example (first set of bars), it looks like 7 samples tested positive at MNRH and 10 samples tested positive at UVRI. This would mean that there were either 3 false-positives at UVRI or 3 false-negatives at MNRH. This is the opposite of what is described in the text. 

Response: Thank you for this keen observation. The confusing graph has been removed and new one plotted as reflected in the revised manuscript. The values were assigned a wrong tittle while inputting.

-As reviewers 1 and 2 noted, the description of dyes and probes in the various real-time PCR assays is confusing and misleading. Dyes are not probes; each assay will use a dye, and some will also use a probe. By placing them in the same pie chart in Figure 2, the authors make it seem as though each assay uses either a dye or a probe.

Response: This section has been extensively revised, we believe your issue has been addressed in the revised manuscript.

-Again, these are just a few examples of where I do not think the authors have adequately addressed prior comments.

Response: We believe the revised manuscript addresses most of your concerns and questions. I would be much excited if the scientific reviewers can clear this paper for publication before it is taken by events.

---

## [Decision Letter · Decision Letter 2]

8 Nov 2023

RE-TESTING AS A METHOD OF IMPLEMENTING EXTERNAL QUALITY ASSESSMENT PROGRAMME FOR COVID-19 REAL TIME PCR TESTING IN UGANDA

PONE-D-23-16243R2

Dear Dr. Okek,

We’re pleased to inform you that your manuscript has been judged scientifically suitable for publication and will be formally accepted for publication once it meets all outstanding technical requirements.

Kind regards,

Chika Kingsley Onwuamah, Ph.D.

Academic Editor

PLOS ONE

Additional Editor Comments (optional):

Reviewers' comments:

Reviewer's Responses to Questions

**Comments to the Author**

1. If the authors have adequately addressed your comments raised in a previous round of review and you feel that this manuscript is now acceptable for publication, you may indicate that here to bypass the “Comments to the Author” section, enter your conflict of interest statement in the “Confidential to Editor” section, and submit your "Accept" recommendation.

Reviewer #1: All comments have been addressed

2. Is the manuscript technically sound, and do the data support the conclusions?

Reviewer #1: Partly

3. Has the statistical analysis been performed appropriately and rigorously? 

Reviewer #1: Yes

4. Have the authors made all data underlying the findings in their manuscript fully available?

Reviewer #1: Yes

5. Is the manuscript presented in an intelligible fashion and written in standard English?

Reviewer #1: Yes

6. Review Comments to the Author

Reviewer #1: I agree to publishing this not because it is novel but because it will provide further information about the quality of testing and preservation of samples collected in another country. Same has been reported in so many other countries globally.

7. PLOS authors have the option to publish the peer review history of their article (what does this mean?). If published, this will include your full peer review and any attached files.

Reviewer #1: No

---

## [Editor Report · Acceptance letter]

3 Jan 2024

PONE-D-23-16243R2 

PLOS ONE

Dear Dr. Okek, 

I'm pleased to inform you that your manuscript has been deemed suitable for publication in PLOS ONE. Congratulations! Your manuscript is now being handed over to our production team.

Kind regards, 

on behalf of

Dr. Chika Kingsley Onwuamah 

Academic Editor

PLOS ONE